# Antimicrobial and Mycotoxin Reducing Properties of Lactic Acid Bacteria and Their Influence on Blood and Feces Parameters of Newborn Calves

**DOI:** 10.3390/ani13213345

**Published:** 2023-10-27

**Authors:** Paulina Zavistanaviciute, Modestas Ruzauskas, Ramunas Antanaitis, Mindaugas Televicius, Vita Lele, Antonello Santini, Elena Bartkiene

**Affiliations:** 1Faculty of Animal Sciences, Institute of Animal Rearing Technologies, Lithuanian University of Health Sciences, Tilzes Str. 18, LT-47181 Kaunas, Lithuania; paulina.zavistanaviciute@lsmu.lt (P.Z.); vita.lele@lsmu.lt (V.L.); 2Department of Food Safety and Quality, Faculty of Veterinary Medicine, Lithuanian University of Health Sciences, Tilzes Str. 18, LT-47181 Kaunas, Lithuania; 3Faculty of Veterinary, Institute of Microbiology and Virology, Lithuanian University of Health Sciences, Mickeviciaus Str. 9, LT-44307 Kaunas, Lithuania; modestas.ruzauskas@lsmu.lt; 4Department of Anatomy and Physiology, Faculty of Veterinary, Lithuanian University of Health Sciences, Tilzes Str. 18, LT-47181 Kaunas, Lithuania; 5Large Animal Clinic, Veterinary Academy, Lithuanian University of Health Sciences, Tilzes Str. 18, LT-47181 Kaunas, Lithuania; ramunas.antanaitis@lsmu.lt (R.A.); mindaugas.televicius@lsmu.lt (M.T.); 6Department of Pharmacy, University of Napoli Federico II, Via D. Montesano 49, 80131 Napoli, Italy; antonello.santini@unina.it

**Keywords:** *Lactiplantobacillus plantarum*, *Lacticaseibacillus paracasei*, antimicrobial properties, mycotoxins, acid whey, newborn calves, blood parameters, feces parameters

## Abstract

**Simple Summary:**

In this study, we hypothesized that the antimicrobial and mycotoxin-reducing properties of lactic acid bacteria (LAB) strains (separate and in combination) can lead to improvements in the health parameters of newborn calves. Considering that, during the first days of life, the formation and domination of the beneficial microbial communities in newborn calves’ digestive systems are very important, the suggested supplementation could lead to a better health status of the animals. In this study, feed supplements were prepared by applying acid whey (AW) for *Lactiplantibacillus plantarum* LUHS135 (L.pl135) and *Lacticaseibacillus paracasei* LUHS244 (L.pc244) biomass growth. The animal trial involved 48 Holstein female calves (12 animals in each group). Control calves were fed with a standard milk replacer, and treated groups (from the 2nd day of life until the 14th day) were supplemented with 50 mL of AW_L.pl135_, AW_L.pc244_, and their combination AW_L.pl135×L.pc244_ (25 mL AW_L.pl135_ + 25 mL AW_L.pc244_) in addition to standard milk replacer. It was found that both the tested strains and their combination inhibited the growth of Gram-positive and Gram-negative pathogens, reduced mycotoxin concentrations in vitro, and were non-resistant to all of the tested antibiotics. The tested supplements showed positive effects on the health parameters of newborn calves, reducing the risk of developing acidosis, decreasing the *Enterobacteriaceae* count, and increasing the LAB count in animal feces. However, more research with larger cohorts of calves is needed to confirm these data and to explain the mechanism of action of the tested supplements.

**Abstract:**

The aim of this study was to evaluate the influence of in acid whey (AW) multiplied *Lactiplantibacillus plantarum* LUHS135 (L.pl135), *Lacticaseibacillus paracasei* LUHS244 (L.pc244), and their biomass combination on newborn calves’ feces and blood parameters. Additionally, the antimicrobial and mycotoxin-reducing properties and the resistance to antibiotics of the tested lactic acid bacteria (LAB) strains were analyzed. In order to ensure effective biomass growth in AW, technological parameters for the supplement preparation were selected. Control calves were fed with a standard milk replacer (SMR) and treated groups (from the 2nd day of life until the 14th day) were supplemented with 50 mL of AW_L.pl135_, AW_L.pc244_, and AW_L.pl135×L.pc244_ (25 mL AW_L.pl135_ + 25 mL AW_L.pc244_) in addition to SMR. It was established that L.pl135 and L.pc244 possess broad antimicrobial activities, are non-resistant to the tested antibiotics, and reduce mycotoxin concentrations in vitro. The optimal duration established for biomass growth was 48 h (LAB count higher than 7.00 log_10_ CFU mL^−1^ was found after 48 h of AW fermentation). It was established that additional feeding of newborn calves with AW_L.pl135_, AW_L.pc244_, and AW_L.pl135×L.pc244_ increased lactobacilli (on average by 7.4%), and AW_L.pl135_ and AW_L.pc244_ reduced the numbers of *Enterobacteriaceae* in calves’ feces. The tested supplements also reduced the lactate concentration (on average, by 42.5%) in calves’ blood. Finally, the tested supplements had a positive influence on certain health parameters of newborn calves; however, further research is needed to validate the mechanisms of the beneficial effects.

## 1. Introduction

Gastrointestinal tract infections and diarrhea are the most important health issues for newborn calves, causing death and huge economic losses [1,2]. However, our previous studies have shown that the addition of supplements based on beneficial microbe biomass to the main feed formula can lead to more effective monitoring of the gastrointestinal microbiota, thus improving the health of farm animals [1,3,4,5,6].

Antibiotic residues, antibiotic-resistant bacteria, and resistance genes are environmental pollutants and are responsible for a tenacious public health crisis worldwide [7]. Antimicrobial substances have long been used to prevent diarrhea in calves; however, researchers are seeking ways to prevent the unnecessary use of antibiotics in food-producing livestock species to limit the development of resistant bacteria [8]. One of the promising alternatives to antibiotics is the use of the antimicrobial properties possessing lactic acid bacteria LAB-based supplements, because of their ability to maintain gastrointestinal microbial balance and enhance immune function by suppressing T helper 2 (Th2)-cell-produced cytokines (IL-4 and IL-5), which increases CD4+ and CD8+ cell counts and levels of IL-12 and INF-γ, and improves phagocytic activity [9,10]. 

There is growing worldwide interest in various LAB strains as antimicrobial supplements in the feed industry; for instance, to reduce the counts of pathogenic bacteria, which leads to a lower incidence of diarrhea in newborn calves [11]. The antimicrobial properties of LAB are pivotal because these starters can be used in feed fermentation and supplement preparation as an alternative to antibiotics, which can reduce the risk of pathogenic and zoonotic bacterial accumulation in the gastrointestinal tract and improve the gut health of livestock [1,7]. Moreover, one of the most important LAB properties is its ability to halt fungal growth and decrease mycotoxin concentrations in the substrate, because contamination with these toxic compounds is a serious problem worldwide [12,13,14]. It is well known that aflatoxin B1 (AFB1) causes liver disorders and, for livestock animals, decreases daily weight gain and leads to lower production efficiency [15,16]. A major concern about AFB1 contamination in feed is that when dairy cows are fed, AFM1 (a toxic metabolite of AFB1) is excreted through their milk, therefore posing risks for human health as well [17]. Fusarium toxins, e.g., zearalenone (ZEA), exhibit strong estrogenic activity, genetic toxicity, hepatotoxicity, and immunotoxicity. The main target is the reproductive system, and such toxins can disturb the gonad and endocrine systems of mammals [15,18,19], causing high economic losses due to abortion, stillbirth, and abnormal fetuses [15,20]. Ochratoxin (OTA) is known for its negative impacts on animal health, causing long-term effects such as liver disorders [21]. It has been reported that LAB can reduce mycotoxin concentrations in two different ways: (I) by enzymatic degradation and (II) by adsorption on the cell surface [13,14,22]. Therefore, LAB strains with high proteolytic, especially enzymatic, activity, such as *Lactobacillus delbrueckii* subsp. *bulgaricus* [23,24], *Lactiplantibacillus plantarum* [24,25], and *Lacticaseibacillus paracasei* [26,27], are recommended for livestock feed additive preparation in order to minimize mycotoxin contamination in feed [14,24]. 

The possible applications of different LAB strains for livestock animal feeding have been studied widely [1,3,4,5,6,28,29]. In this study, we hypothesized that the antimicrobial and mycotoxin-reducing properties of LAB strains (separate and in combination) can lead to an improvement in the health parameters of newborn calves. Considering that during the first days of life, the formation and domination of beneficial microbial communities in newborns calves’ digestive systems are very important, the suggested supplementation can lead to a better health status of the animals.

*Escherichia coli* (VTEC), also known as Shiga-toxin-producing is a zoonotic pathogen, causing serious public health problems, for instance outbreaks, bloody diarrhea and hemolytic uremic syndrome, a leading cause of acute renal failure in children, etc. [30,31,32]. It is well known, that ruminants are the main reservoirs of *E. coli* O157, and through their feces, these pathogens can contaminate drinking water and food, as well as through contact with farm animals and human-to-human transmission, which can cause serious health problems or even outbreaks [30,31]. However, the prevalence of *E. coli* O157 in the feces of beef cattle can be reduced by direct feeding of LAB [30,33,34].

However, the application of LAB biomass as a feed supplement is limited due to the expensive commercial media used for bacterial biomass growth. For this reason, new alternatives for LAB biomass production, in a cheaper and more sustainable manner, are being sought. The food industry generates high quantities of by-products [35,36], which can be used for microorganism biomass preparation on an industrial scale. The dairy industry by-product, whey, can be acidic or sweet depending on casein precipitation. It has been estimated that whey production worldwide is around 160 million tons per year and only 30–50% of this amount is used efficiently [37,38]. It is known that whey has an undesirable environmental impact on soil and water due to its high mineral and lactose concentrations, and it is considered to be a potential environmental pollutant [37,39]. The valorization of AW is limited due to its high acidity; however, it could be effectively applied in supplements for animal nutrition preparation [40]. It has been reported that AW can be applied for LAB biomass growth and encapsulation. This is also very important, as LAB are highly sensitive to environmental conditions [41]. In this study, in order to ensure effective biomass growth in AW, the technological parameters for feed supplement preparation were optimized. 

The aim of this study was to evaluate the influence of in AW multiplied *Lactiplantibacillus plantarum* LUHS135 (L.pl135), *Lacticaseibacillus paracasei* LUHS244 (L.pc244), and their biomass combination on newborn calves’ feces and blood parameters. Additionally, the antimicrobial and mycotoxin-reducing properties and resistance to antibiotics of the tested LAB strains were analyzed. In order to ensure effective biomass growth in AW, technological parameters for the supplement preparation were selected. 

## 2. Materials and Methods

### 2.1. Materials Used in Experiment

*Lactocaseibacillus paracasei* LUHS244 (L.pc244) and *Lactiplantibacillus plantarum* LUHS135 (L.pl135) strains were isolated from spontaneously fermented cereals [42]. Strains before the experiment were stored at –80 °C (*PRO-LAB Diagnostics*, Bromborough, United Kingdom), supplemented with 20% glycerol (Sigma Aldrich, Darmstadt, Germany). Before the experiment, LAB strains were propagated in MRS broth (CM 0359, Oxoid Ltd., Hampshire, United Kingdom) at 30 °C for 48 h.

AW (lactose 4.0%, protein 0.8%, lactic acid 0.5%, minerals 0.6%, total solids 6.5%) is the by-product resulting from cottage cheese production and was obtained from JSC “Pieno zvaigzdes” (Kaunas, Lithuania) and stored at −18 °C until use in L.pc244 and L.pl135 feed supplements for newborn calves’ feed preparation. 

Pathogenic and opportunistic strains (Pseudomona aeruginosa 17–331, Staphylococcus aureus M87fox, *Escherichia coli* (hemolytic), Streptococcus mutans, Enterococcus faecium 103, Klebsiella pneumoniae, Enterococcus faecalis 86, Bacillus cereus 18 01, Proteus mirabilis, Citrobacter freundii and Salmonella enterica 24 SPn06), used for evaluation of the antimicrobial activity of L.pl135 and L.pc244, were obtained from the Lithuanian University of Health Sciences (Kaunas, Lithuania) collection. In the experiments we used pathogenic and opportunistic wild bacterial strains that were previously isolated from humans and animals.

Gentamicin (CN), tetracycline (TE), erythromycin (E), amoxicillin (AML), and trimethoprim (TMP) MIC test strips were purchased from “Liofilchem” (Roseto degli Abruzzi, Italy) and used for the antibiotic resistance evaluation of L.pl135 and L.pc244 strains.

AFB1, OTA, ZEA, T-2 toxin (T-2), and HT-2 toxin (HT-2) standards with certifications and verification of the contents of the compounds were acquired from Romer Labs Biopure (Tulln, Austria). Prior to use, the abovementioned mycotoxin standards were rendered in acetonitrile in order to make stock solutions with concentrations of 1 mg mL^−1^. The prepared solutions were kept at −20 °C for a restricted period, but not for longer than 6 months, as per the manufacturer’s recommendations.

### 2.2. Principal Scheme of the Experiment

First of all, the antimicrobial activities of L.pl135, L.pc244, and their combination (L.pl135 × L.pc244) against pathogenic and opportunistic strains (mentioned in Section 2.1) and resistance to antibiotics (mentioned in Section 2.1) were evaluated (Figure 1). In addition, both LAB strains (L.pl135 and L.pc244) were tested for mycotoxin reduction properties. For this purpose, different concentrations of mycotoxin solutions (AFB1 concentrations of 2 μg L^−1^ and 10 μg L^−1^, OTA concentrations of 3 μg L^−1^ and 10 μg L^−1^, ZEA concentrations of 75 μg L^−1^ and 350 μg L^−1^, T-2 concentrations of 50 μg L^−1^ and 1000 μg L^−1^, and HT-2 concentrations of 50 μg L^−1^ and 1000 μg L^−1^) were prepared and two different concentrations of L.pl135 and L.pc244 were added (3 and 7%). The mycotoxin-reducing properties of the LAB strains were evaluated by testing mycotoxin concentrations in prepared mixtures after 0, 1, 3, and 6 h of incubation. Antimicrobial activity and mycotoxin reduction properties were tested in vitro.

During the second stage (Figure 1), LAB strains were multiplied (separately) in sterilized AW for 12, 24, 36, and 48 h in order to select the most appropriate fermentation duration (according to the highest viable LAB cell count and acidity parameters: pH, titratable acidity (TA) and total lactic acid, L(+) lactic acid isomer and D(−) lactic acid isomer concentration) for supplement preparation. During the third stage of the experiment, the prepared supplements were used for feeding newborn calves. The feeding experiment is described in detail in Section 2.8. In order to determine the influence of the prepared feed supplements on newborn calves’ health parameters, the blood gas, biochemical and feces’ microbial parameters of the calves (before and after the experiment) were analyzed.

### 2.3. Evaluation of Antimicrobial Activities of Lactic Acid Bacteria Strains

The antimicrobial activity of LAB strains was tested via an agar well diffusion assay [42]. For this purpose, a 0.5 McFarland Unit density suspension (~10^8^ log_10_ CFU mL^−1^) of each pathogenic bacterial strain was inoculated onto the surface of cooled Mueller–Hinton Agar (Oxoid, UK) using sterile cotton swabs. Wells of 6 mm in diameter were punched in the agar and filled with 50 µL of L.pl135, L.pc244, and L.pl135 × L.pc244. The antimicrobial activities against the tested bacteria were determined by measuring the diameter of the inhibition zones (mm).

### 2.4. Evaluation of Lactic Acid Bacteria Antibiotic Resistance 

The minimum inhibitory concentrations (MICs) of GEN, TET, ERY, AML, and TMP were determined using the micro-dilution method [41]. The MICs were evaluated as the lowest concentrations of given antibiotics at which no growth of the test organism was observed. Microbiological cut-off values were used as the interpretative criteria for susceptibility testing, according to the EFSA-FEEDAP guidelines [43].

### 2.5. Analysis of Lactic Acid Bacteria Mycotoxin-Reducing Properties

Prior to the experiment, L.pl135 and L.pc244 strains were multiplied as described in Section 2.1 into 10 mL of fresh MRS broth, two different concentrations (3 and 7% (*v/v*)) of L.pl135 and L.pc244 (viable cell count ~10^8^ CFU mL^−1^) were added separately, and these were spiked with two concentrations of tested mycotoxins: (I) 2 μg L^−1^ and 10 μg L^−1^ of AFB1; (II) 3 μg L^−1^ and 10 μg L^−1^ of OTA; (III) 75 μg L^−1^ and 350 μg L^−1^ of ZEA; (IV) 50 μg L^−1^ and 1000 μg L^−1^ of T-2; (V) 50 μg L^−1^ and 1000 μg L^−1^ of HT-2. Each sample, with different LAB strains, different LAB concentrations, and different mycotoxin concentrations, was incubated at 30 °C for 6 h. In order to evaluate the changes in mycotoxin concentrations in samples, analyses were performed before the experiment and after 1, 3, and 6 h of incubation. Results are illustrated with HemI 2.0: an online service for heatmap illustration program [44].

Samples were centrifuged at 3044× *g* (Centrifuge 5810R, Eppendorf, Wesseling-Berzdorf, Germany) for 5 min at 4 °C to separate the cells. A single-stage extraction method, followed by ultra-high-performance liquid chromatography on an Acquity UPLC instrument (Waters, Milford, MA, USA) coupled to a QTrap 5500 (AB SCIEX, Framingham, MA, USA) triple quadrupole tandem mass spectrometer, equipped with an electrospray ion source, was used for the analysis of mycotoxins in the samples, according to the procedure described by Reinholds at al. [45], and details are given in Appendix A. Three parallel replicates of each sample were obtained.

### 2.6. Fermentation and Analysis of Acid Whey Parameters 

Before fermentation, AW was autoclaved for 15 min at 121 °C, and cooled to 30 °C. The content of 40 ± 1 mL working volume, was filtered through a 1 mm diameter mesh filter into borosilicate glass (Duran) bottles (50 mL). The process was performed in a laminar chamber. Then, 3% (*v/v*) of each LAB strain was added to the AW and fermentation was performed under anaerobic conditions with a modified carbon dioxide atmosphere in a chamber incubator (Memmert GmbH + Co. KG, Schwabach, Germany) for 48 h at 30 °C. For experiments, separate samples were prepared, in order not to damage anaerobic conditions. In vitro research (lactic acid bacteria count and acidity parameters) was evaluated from the same sample.

In order to optimize the biomass growth technology, samples were tested after 12, 24, 36, and 48 h of fermentation. For viable LAB number evaluation, the serial dilution method was applied as described in ISO 15214:1998 [46] and Appendix A.

The pH values of fermented AW were determined with a pH electrode (PP-15, Sartorius, Goettingen, Germany) [47].

The titratable acidity (TA) was evaluated using 10 mL of the sample mixed with 90 mL of distilled water [48].

L(+) and D(−) lactic acid isomers and total lactic acid concentrations were analyzed with a specific Megazyme assay kit (Megazyme Int., Bray, Ireland), according to the manufacturer’s instructions [49]. 

### 2.7. Feeding Experiment Design

A total of 48 female Holsteins calves were randomly divided into 4 homogeneous experimental groups, each containing 12 animals, according to body weight (34.00 ± 3 kg) from the second day of life and the experiment was carried out till the 14th day of life. The control group (C_control)_ was fed with a SMR containing skimmed milk powder, whey powder, hydrolyzed wheat protein, minerals and vitamins (analytical composition: protein—22.0%, fat—17.0%, fiber—0.0%, minerals—9% (Se, Cu, Fe) and vitamins (A, E, D3) on a dry matter basis) diet; the treated groups were fed with the same diet supplemented with 50 mL of AW fermented with L.pl135 (C-AW_L.pl135_), 50 mL of AW fermented with L.pc244 (C-AW_Lpc244_), and 50 mL of AW fermented with L.pl13 and L.pc244 (C-AW_L.pl135×Lpc244_). The LAB count in the AW substrate was higher than 7.0 log_10_ CFU mL^−1^ (Table 1). All prepared feed supplements (AW_L.pl135_, AW_L.pc244_, and AW_L.pl135×Lpc244_) were inserted and stirred in the milk replacer. The SMR powder (130 g L^−1^) was reconstituted in 55 °C water and calves were fed from a bucket during the morning and afternoon, while the feed temperature was 39 °C. Each calf was placed in an individual indoor box (2.00 × 1.25 m), with free access to water. Calves were fed individually twice a day (7.00 a.m. and 1 p.m.) with non-medicated SMR (8–10 L per calf per day). 

Health parameters: physiological indices of the blood (pH; pCO_2_; pO_2_; O_2_ saturation; Na; K; iCa; tCO_2_; Hct, Hb; Glu; lactates) were investigated before and after the experiment (on the 2nd and 14th days of life) according to the procedures described by Zavistanaviciute et al. [4] and are given in Appendix A. Fecal microbial parameters (counts of LAB, total number of cultivable bacteria, *Enterobacteriaceae*, and the number of yeasts and molds) were analyzed according to the procedures described in ISO 15214:1998 [46], ISO 4833-2:2013 [50], ISO 21528-2:2017 [51], and ISO 21527-2:2008 [52], respectively. The methods are also described in detail in Appendix A.

### 2.8. In Vivo Experiments’ Ethical Guidelines

The animals were hosted indoors, being individually tethered, and cared for in accordance with the Lithuanian State Food and Veterinary Service Requirements. Research was carried out in accordance with the 6 November 1997 Republic of Lithuania act covering animal care and maintenance, and in accordance with the appropriate legal act, i.e., 8-500, Official Gazette, No 130-6595: 2012 [53].

### 2.9. Statistical Analysis

All assays were carried out in triplicate, and the results are expressed as the mean ± standard error (SE). The results were analyzed using the SPSS statistical package for Windows V27.0 (SPSS Inc., Chicago, IL, USA, 2020). In order to evaluate the influence of selected variables, a multivariate analysis of variance (ANOVA) and the Tukey HSD test (as a post hoc test) were performed. To examine the correlations between AW sample acidity parameters, Pearson correlations (r) were calculated. The results were considered to be statistically significant at *p* ≤ 0.05.

## 3. Results and Discussion 

### 3.1. The Tested Lactic Acid Bacteria Strains’ Antimicrobial Characteristics and Resistance to Antibiotics 

The results of the L.pl135 and L.pc244 antimicrobial activity against the tested pathogenic and opportunistic strains and their resistance to antibiotics are presented in Table 2 and Table 3, respectively. L.pl135 and L.pc244 inhibited the growth of all the tested pathogenic and opportunistic strains: *P. aeruginosa*, *S. aureus*, *E. coli*, *S. mutans*, *E. faecium*, *K. pneumoniae*, *E. faecalis*, *B. cereus*, *P. mirabilis*, *C. freundii*, and *S. enterica.* The combination of L.pl135 and L.pc244 inhibited 10 out of 11 tested pathogens, except for *K. pneumoniae*, with the diameter of the inhibition zone from 9.0 to 17.8 mm (for *E. coli* and *C. freundii*, respectively). The largest diameter of the inhibition zone for L.pl135 was observed against *B. cereus* (17.0 mm). The largest diameters of the inhibition zones for the L.pc244 and the L.pl135 × L.pc244 combination were observed against *C. freundii* (16.0 and 17.8 mm, respectively). The smallest diameters of the inhibition zone were found against *E. coli* (10.5 mm for both the tested strains) and *S. enterica* (10.5 mm for both the tested strains). The same tendency was established for the L.pl135 × L.pc244 combination against *E. coli* (9.0 mm). 

*Lp. plantarum* is one of the most important strains of *Lactobacilli* spp. [54,55]. The biosynthetic production of bioactive peptides, enzyme systems, organic acids, exopolysaccharides, and vitamins is considered the key mechanism of antioxidant, antimicrobial, and probiotic activities [55]. Our study results are in agreement with those of other studies, which have reported that *Lp. plantarum* has strong antimicrobial activity against Gram-positive pathogenic strains, including *S. aureus* [42,55], *L. monocytogenes* [55], and *E. feacalis* [42,56]. Furthermore, it has been reported that the *Lp. plantarum* strain inhibits the growth of Gram-negative pathogenic strains, i.e., *E. coli* [57,58], S. *typhimurium* [55,59], and *P. aerugonosa* [60,61]. In addition, the *Lc. paracasei* strain has strong potential as an antimicrobial supplement, due to its ability to inhibit the growth of a broad spectrum of pathogenic strains, including *K. pneumoniae* [42,61], *A. baumannii* [42,61], *E. coli* [62], *Salmonella enteritidis*, and *Enterococcus* spp. [63]. These results can be explained by the LAB strain’s ability to produce protease-sensitive bacteriocins, as well as hydrogen peroxide and organic acids (lactic, acetic, and propionic acids), which obstruct the growth of pathogens [64,65]. 

Recent studies have shown that LAB strains isolated from spontaneously fermented cereals are more susceptible to GEN, TET, and ERY [4,29,41] in comparison with strains isolated from milk products [62]. According to Reuben et al. [62], 100% of *Lc. casei*, *Lp. plantarum*, *Limosilactobacillus fermentum*, and *Lc. paracasei* isolated from cow’s milk showed resistance to oxacillin, 75% of isolates were resistant to vancomycin and streptomycin, and 50% were resistant to ERY and chloramphenicol. These results can be explained by the fact that antibiotics are widely used in food-producing animals, and this contributes to the emergence of antibiotic-resistant bacteria present in the intestinal microflora [66]. Antibiotic-resistant bacterial strains can carry resistance factors to pathogens through the exchange of genetic material [67]. One of the most important safety considerations for LAB implementation in feed/food production is testing that a potential probiotic strain does not carry transferable resistance genes [68]. According to our results, one of the alternatives could be the application of LAB strains isolated from plant-based substrates, such as fermented cereal, due to their higher susceptibility to antibiotics, however, further studies should be performed, in order to analyze resistance genes of tested LAB strains.

### 3.2. Lactic Acid Bacteria Mycotoxin Reduction Properties 

For the mycotoxins AFB1, OTA, ZEA, T-2, and HT-2, the reduction properties of the L.pl135 and L.pc244 strains are shown in Figure 2. The 3% concentration of L.pl135 reduced the AFB1 concentration in 2 μg L^−1^ and 10 μg L^−1^ samples. After 3 h of incubation with 3% L.pl135, the highest reduction in AFB1 in the 2 μg L^−1^ concentration samples was found (40.09 ± 0.28%, *p* < 0.005). However, the reduction in AFB1 for the 10 μg L^−1^ concentration samples at the same L.pl135 concentration was lower on average by 70 %, and the highest reduction was shown after 1 h of incubation (8.89 ± 0.14%, *p* ≤ 0.05). Similar tendencies in the AFB1 concentration reduction at 7% for L.pl135 were established. After 1 h of incubation with the 7% L.pl135 strain, the highest reductions for both the 2.0 μg L^−1^ and 10 μg L^−1^ AFB1 concentration samples were found (by 50.0 ± 0.27 and 11.1 ± 0.24%, *p* < 0.004, and *p* ≤ 0.05), respectively). Also, the reduction in AFB1 concentration in 10 μg L^−1^ samples, after their treatment with 3 and 7% L.pl135, showed a tendency to decrease the AFB1 concentration. However, the highest mycotoxin concentration reduction was established for 7% L.pl135 in both 2.0 μg L^−1^ and 10 μg L^−1^ AFB1 concentration samples. The 3% L.pc244 strain did not show a reduction capability of AFB1 in 2.0 μg L^−1^ concentration samples after 1, 3, and 6 h of incubation. After 3 h of incubation with the 3 and 7% L.pc244 strain, the highest (7.45 ± 0.22 and 8.79 ± 0.26%, *p* ≤ 0.05, and *p* ≤ 0.04, respectively) reductions for AFB1 in 10 μg L^−1^ concentration samples were found. In comparison, for the AFB1 reduction in 2.0 μg L^−1^ concentration samples, it was established that the duration of incubation was not a significant factor in respect of the AFB1 concentration for 7% L.pc244 strain-treated samples, and in all samples, an average AFB1 reduction of 28.6% was detected.

One of the most important properties of LAB is its ability to reduce mycotoxin concentrations in food and feed [69]. It has been reported that AFB1 concentration reduction is LAB-strain-specific [70,71,72,73,74], and the reduction of this toxic compound can be explained by adsorption based on van der Waals, hydrophobic, electrostatic, and hydrogenic bond interactions [70,71,75,76]. According to other studies, *Lp. plantarum* can bind from 21 to 60% of AFB1 after 30 min of incubation [70,77]. According to Zokaityte et al. [78], fermentation with *Lc. paracasei* and *Lc. casei* strains can lead to lower AFB1 content in wheat bran (on average, by 52 and 46%, respectively).

Our results show that, after 3 and 6 h of incubation with 3% L.pl135, the reduction in OTA in 10 μg L^−1^ concentration samples was, on average, 11.2% (*p* < 0.02) (Figure 2). In comparison, for samples treated with 3 and 7% L.pl135, the latter L.pl135 concentration showed lower OTA reduction properties in 10 μg L^−1^ concentration samples (on average, a 48.5% (*p* < 0.02) reduction was found). Only 7% L.pl135 after 6 h of incubation in 3 μg L^−1^ OTA concentration samples was capable of reducing this mycotoxin concentration on average by 5.88% (*p* < 0.01). The L.pc244 strain, at 3% concentration after 6 h of incubation in 3 μg L^−1^ OTA concentration samples, diminished this mycotoxin concentration on average by 6.25% (*p* < 0.04). However, after 1 and 3 h of incubation, the concentration of OTA in samples was not reduced. However, 3% L.pc244 in 10 μg L^−1^ OTA samples, after 1 and 6 h of incubation, reduced the OTA concentration on average by 7.23 (*p* < 0.04) and 8.43% (*p* < 0.02), respectively. A slightly lower reduction in OTA in the 3 μg L^−1^ concentration samples by the L.pc244 strain after 1 and 6 h of treatment was found (on average, 5.88%, *p* < 0.02). 

OTA is a secondary metabolite produced by *Aspergillus* and *Penicillium* species, and this toxin is second in importance in feed after AFB1 [79,80]. According to Luz et al. [14], LAB has a different OTA reduction capacity, which depends on substrate acidity and strain characteristics. For instance, OTA reduction by the *Lp. plantarum* strain at pH 3.5 can vary from 30 to 97%, while at pH 6.5, the range of reduction can be from 90 to 97% [14]. It has been reported that OTA reduction by the *Lc. paracasei* strain can range from 21 to 16% when substrate pH values are 3.5 and 6.5, respectively [14]. Zokaityte et al. [78] reported that OTA reduction in wheat bran fermented with *Lp. paracasei* was 64% in comparison with non-treated samples.

After 6 h of incubation with L.pl135, the reduction in ZEA in the 75 and 350 μg L^−1^ concentration samples was 10.4 (*p* < 0.01) and 11.3% (*p* < 0.03), respectively (Figure 2). However, the reduction in ZEA at both sample concentrations, after 1 and 3 h of incubation with 3% L.pl135, was not observed. However, the 7% L.pl135 concentration after 1, 3, and 6 h of incubation decreased the ZEA content in both (75 and 350 μg L^−1^) tested samples. The highest reduction capability was shown for 7% L.pl135 in the 350 μg L^−1^ concentration of ZEA samples after 1 h of incubation (on average, a 16.1% (*p* < 0.02) lower ZEA content was found). In the 75 μg L^−1^ concentration samples incubated with the 7% L.pl135 strain for 6 h, ZEA reduction was, on average, 17.8% (*p* < 0.02). After 6 h of incubation with 3% of L.pc244, the reductions in ZEA in the 75 and 350 μg L^−1^ concentration samples were on average, 8.95 (*p* < 0.03) and 13.20% (*p* < 0.03), respectively. After 1 h of treatment with the same LAB strain concentration, lower reductions in ZEA in the 75 and 350 μg L^−1^ concentration samples were established (on average, 4.95 (*p* < 0.01) and 5.86% (*p* < 0.005), respectively), in comparison with 6 h treated samples. However, the reduction in ZEA in both tested concentration samples after 3 h of treatment was not observed. The L.pc244 strain, at a 7% concentration, decreased the ZEA only in the 75 μg L^−1^ concentration samples (after 1, 3, and 6 h of treatment, on average, by 3.05 (*p* < 0.03), 5.95 (*p* < 0.01), and 15.8% (*p* < 0.02), respectively). In the 350 μg L^−1^ concentration samples, treated with the 7% L.pc244 strain, a reduction in ZEA was not established. 

The reduction in ZEA is strain-specific and depends on the LAB cell wall protein type, as well as the structure, which is distinct for different strains [81]. According to Złoch et al. [82], ZEA reduction by the *Lc. paracasei* strain involves two stages: (I) biosorption/binding of mycotoxins in bacterial cells and (II) metabolization of ZEA to less toxic α-ZOL and β-ZOL forms. Adunphatcharaphon et al. [81] reported that adsorption is a mechanism involved in ZEA reduction by plant-derived *Lp. plantarum,* and reduction in this mycotoxin concentration can vary from 0.5 to 21%. 

We found that 3% L.pl135 reduced T-2 concentration in 1000 μg L^−1^ samples (on average, by 24.8%, *p* < 00007) after 1, 3, and 6 h of incubation (Figure 2). Similar tendencies of T-2 mycotoxin reduction capability by the 7% L.pc244 strain were found. The reduction in T-2 for 1000 μg L^−1^ concentration samples by the 7% L.pc244 strain after 1, 3, and 6 h of incubation was, on average, 19.7 (*p* < 0.02), 26.0 (*p* < 0.008), and 24.0% (*p* < 0.008), respectively. However, neither of the tested L.pl135 concentrations were effective for T-2 reduction in the 50 μg L^−1^ concentration samples. The highest reduction in T-2 by 3% L.pc244 in the 50 μg L^−1^ concentration T-2 samples after 3 h of incubation was found (on average, by 52.3%, *p* < 0.0001). The highest T-2 concentration reduction was shown for the 3% L.pc244 in 1000 μg L^−1^ concentration T-2 samples (on average, reduction after 1 h of incubation was 17.9%, *p* < 0.02). By increasing the L.pc244 concentration to 7%, the T-2 concentration in the 50 μg L^−1^ samples after 6 h of incubation was reduced on average by 38.8% (*p* < 0.005). However, the highest reduction was found by using the 7% L.pc244 strain in the 1000 μg L^−1^ concentration T-2 samples after 3 h of incubation (on average, a 22.5% (*p* < 0.009) lower T-2 concentration was established). 

Our findings are in agreement with Zokaityte et al.’s [78] published results, which suggest that *Lp. paracasei* reduces the T-2 toxin concentration in the cereal outer layer. It was reported that LAB strains have a relationship with the 12,13-epoxy ring that is crucial for the T-2 concentration reduction [83]. T-2 can also be metabolized to HT-2, and the toxicity of T-2 might be partially attributed to HT-2 [84].

In comparison, for HT-2 concentrations, only after 3 h of incubation with the 3 and 7% L.pl135 strain was the reduction in HT-2 in the 1000 μg L^−1^ concentration samples established (on average, by 5.35 (*p* < 0.02) and 2.96% (*p* < 0.01), respectively) (Figure 2). In addition, the 3% L.pc244 strain reduced the HT-2 concentration in the 50 μg L^−1^ samples (on average, by 40.7 (*p* < 0.007), 34.8 (*p* < 0.006), and 37.9% (*p* < 0.003) after 1, 3, and 6h of treatment, respectively). However, by using 7% L.pc244, a reduction in HT-2 was found after only 3 h of incubation in the 1000 μg L^−1^ concentration HT-2 samples (on average, a 3.34% (*p* ≤ 0.05) lower HT-2 concentration was found). 

Type A trichothecenes (for instance, T-2 toxin, HT-2 toxin, or diacetoxyscirpenol) are more toxic for animals and humans than other foodborne trichothecenes such as the type B group (deoxynivalenol, nivalenol) [85]. According to Juodeikienė et al. [86], degradation of trichothecenes by LAB strains depends on the LAB strain used, as different strains may have different cell densities and viabilities in contaminated foods and feeds with diverse pH values of the media. Trichothecene concentrations in contaminated media also play a significant role during the detoxifying process [86,87]. It is important to eliminate HT-2 in livestock feed due to its ability to trigger neural disturbances, vomiting, and feed intake reduction, as well as decreasing productivity [88]. Moreover, T-2 can be rapidly metabolized to HT-2 toxin in the intestinal tract, and it is known that both mycotoxins in vivo are within a similar range of toxicity [87,88]. 

Our findings are in agreement with other published results [13,78,86,87], which suggest that the type of LAB strain, concentration of LAB strain used, mycotoxin concentration in the solution, and duration of incubation are significant (*p* < 0.0001) factors in terms of the AFB1, OTA, ZEA, T-2, and HT-2 concentrations in samples. Moreover, the abovementioned factors’ interactions were significant (*p* < 0.0001) in respect of the AFB1 and ZEA concentrations in samples. Finally, 3% concentrations of both tested LAB strains, as well as their combination, could be recommended as a natural and environmentally friendly approach for mycotoxin reduction, because the combination of L.pl135 and L.pc244 strains could reduce AFB1, OTA, ZEA, T-2, and HT-2 concentrations. 

### 3.3. Changes in Lactic Acid Bacteria Count and Acidity Parameters of Acid Whey during Fermentation

The LAB counts and acidity parameters of fermented AW are presented in Table 4. It was found that in both AW samples, fermented with L.pl135 and L.pc244 strains, the LAB count increased with an increasing duration of fermentation. After 24 h of fermentation, the LAB counts in AW_L.pl135_ and AW_L.pc244_ samples, 6.89 ± 0.05, and 6.07 ± 0.12 log_10_ CFU mL^−1^, respectively, were found. Similar tendencies were found after 36 and 48 h of AW fermentation with both LAB strains. The highest viable LAB cell counts were found in AW_L.pl135_ and AW_L.pc244_ samples after 48 h of fermentation (on average, 7.51 log_10_ CFU mL^−1^). In addition, by reducing samples’ pH, the LAB count was increased, and a very strong negative correlation (r = −0.946, *p* < 0.003) between LAB count and sample pH was found. One of the major metabolites generated by LAB during lactic fermentation is lactic acid [89]. Also, our previous studies showed, that, L.pl135 and L.pc244 can metabolize lactose, as well as tolerate low pH values [42]. These characteristics are important if we choose alternative substrates, especially dairy industry by-products with high acidity, for LAB multiplication. 

We compared pH values of samples fermented with different LAB strains (Table 4). Lower pH values were shown by samples fermented with the L.pl135 strain in comparison with the L.pc244 strain. The lowest pH values were shown by 48 h fermented samples (AW_L.pl135_ and AW_L.pc244_ samples’ pH values were, on average, 3.89 and 4.03, respectively). The latter samples had the highest TA values (7.80 and 6.90° N, respectively). These findings are in agreement with Zokaityte et al. [90], who reported that for dairy industry by-products such as milk permeate, fermentation with *lactobacilli* leads to viable LAB cell counts higher than 8.00 log_10_ CFU mL^−1^ and substrate pH values lower than 4.00 after 48 h of fermentation. Our previous studies showed that *Lacticaseibacillus casei* and *Liquorilactobacillus uvarum* strains multiplied in AW for 48 h led to LAB counts higher that 8.50 log_10_ CFU mL^−1^ in AW samples, while pH values varied from 3.61 to 3.74 [47]. It is also known that various lactobacilli strains tolerate and grow in harsh conditions; for instance, during transit in the gastrointestinal tract, they can survive at low pH (from 2.0 to 4.0) and are resistant to bile salts as well as pancreatic juice exposure in the intestinal tract [91,92]. These characteristics of LAB are highly desirable for the production of feed or feed supplements. 

Furthermore, we determined that total lactic acid, as well as L(+) lactic acid and D(−) lactic acid concentrations, increased in fermented AW (Figure 3). L(+) lactic acid concentration, after 24, 36, and 48 h of fermentation, increased (on average, by 1.95, 3.32, and 3.80 times, respectively) in AW_L.pc244_ samples in comparison with 12 h fermented AW. Similar tendencies of L(+) lactic acid concentration were observed in AW_L.pl135_ samples. It was established that L(+) lactic acid concentrations in AW-L.pl135 samples were 1.82, 2.36, and 2.88 times higher after 24, 36, and 48 h of fermentation, respectively, in comparison with 12 h fermented AW. In all cases, on average, 68.3% higher D(−) lactic acid concentrations were found in AW_L.pl135_ samples, in comparison with AW_L.pc244_ samples. The highest concentration of L(+) lactic acid was found in 48 h fermented AW_L.pc244_ samples (5.97 g L^−1^). After 48 h of fermentation, on average, an 18.4% lower concentration of L(+) lactic acid in AW_L.pl135_ samples was found in comparison with L(+) lactic acid concentrations in AW_L.pc244_ samples after the same fermentation duration. In addition, on average, a 56.5% lower concentration of D(−) lactic acid was found in 48 h fermented AW_L.pc244_ samples in comparison with AW_L.pl135_ after the same fermentation duration. 

LAB strains during fermentation have a tendency to have a post-acidification effect on the substrate, which results in higher TA and lower pH values [90,93]. LAB also metabolizes carbohydrates into organic acids (in our case, lactose from AW to L(+) lactic acid and D(−) lactic acid isomers), which increases the acidity of the substrate [47,94,95]. Production of L(+) lactic acid isomer is desirable; however, D(−) lactic acid isomer is harmful and excessive intake can have fatal consequences [96,97]. Our study results are in accordance with Klupsaite et al.’s [98] published results, which show that the total lactic acid concentration in whey fermented for 48 h with *Lactobacillus delbrueckii* subsp. *bulgaricus* was 5.28 g L^−1^; we determined, on average, 6.17 g L^−1^ quantities in AW. LAB can produce one isomeric form of lactic acid or a mixture of both lactic acid isomers [98,99], and in our case, both D(−) and L(+) lactic acid isomers were found. However, L(+) lactic acid isomers prevailed in samples fermented with L.pl135 and L.pc244 strains. Two-way ANOVA analyses showed that LAB strain and duration of fermentation, as well as the interaction of these factors was significant (*p* < 0.001) in respect of L(+) and D(−) lactic acid isomer concentrations in fermented AW. 

### 3.4. The Influence of AW_L.pl135_, AW_L.pc244_ and Their Combination on Newborn Calves’ Blood Parameters

Dietary supplementation of AW_Lpl135_, AW_Lc244_, and AW_L.pl135×L.pc244_ had a clear impact on the hematological profile of newborn calves (Table 5). It was found that the concentration of lactate in the C-AW_L.pl135_, C-AW_L.pc244_, and C-AW_L.pl135×L.pc244_ groups’ blood was significantly diminished (on average, by 42.1, 31.4, and 54.1%, respectively), in comparison with the C_control_ group (reduction by 13.3% was observed). These results are prospective, and a lower concentration of lactate in the blood can prevent acidosis in newborn calves. It is essential that in calves’ feed supplements prepared from fermented LAB materials, the main lactic acid isomer is L(+). According to Bartkiene et al. [100], if LAB produces both isomers of lactic acid, it is relevant to evaluate the L(+)/D(−) lactic acid isomer ratio. It is known that mammalian cells solely generate the L(+) lactic acid isomer, whereas they only contain L-lactate dehydrogenase and can metabolize considerable concentrations of L(+)-lactic acid isomer [100,101,102]. However, D(−) lactic acid isomer can accumulate since it is not degraded and can cause acidosis [101]. In addition, D(−) lactic acid isomer can be generated during carbohydrate metabolism by bacteria in the intestinal tract [103,104]. For this reason, it is important to control lactate in calves’ blood, and eligible feed supplements could prevent acidosis.

Additionally, a significant reduction in the pCO_2_ concentration in the C-AW_L.pl135_ and C-AW_L.pc244_ groups’ blood (on average, by 54.7 and 37.7%, respectively) was found, while in the C_control_ group, reduction by 14.0% was observed. Conversely, a rise in pO_2_ concentration in the C-AW_L.pl135_, C-AW_L.pc244_, and C-AW_L.pl135 × L.pc244_ groups was found (on average, by 8.80, 44.0, and 36.0%, respectively), while in the C_control_ group this parameter was reduced, on average, by 1.5%, in comparison with the results before treatment. After 14 days of treatment, the Hgb concentration in the C-AW_L.pc244_ and C-AW_L.pl135 × L.pc244_ groups’ blood increased, on average, by 14.0 and 27.2%, respectively, in comparison with the C_control_ group, in which Hgb decreased on average by 17.4%. The higher concentration of Hgb in blood could be associated with a lower risk of anemia in newborn calves. The results of the ANOVA test indicated that there is a significant effect (*p* < 0.0001) of the treatment duration, as well as the interaction of the treatment duration and the supplement used, on most of the tested blood parameters of calves (with the exception of pH, O_2_ saturation, Na, tCO_2_, Hct fraction, and Glu concentrations). 

Functional properties of probiotic LAB, such as tolerance to gastrointestinal tract conditions, adhesion to the intestinal epithelium, immunomodulation, and bacterial antagonism are the most important components of the probiotic potential, and have a positive influence on the health parameters of the host [105,106,107]. However, sensitive blood parameters are also important. The *Lp. plantarum* (*Lplant*-B80-like) strain is likely endemic in farms and well-adapted to the intestinal tract of neonatal calves [108], and the *Lp. plantarum* ACA-DC 2640 strain increased total IgG serum levels [107]. It has been reported that a feed supplement containing *Liq. uvarum* LUHS245 can reduce the risk of calves developing acidosis and the serum alanine aminotransferase concentration in newborn calves’ blood (50 mL per day per 14 days) [4]. Lactate plays a significant role in neonatal acidosis and subsequent asphyxia, as well as being responsible for metabolic acidosis, and persists in the blood in increased concentrations for considerably longer than CO_2_ [109]. Data regarding the influence of *Lc. paracasei* on newborn calves’ blood profiles are scarce; however, it has been reported that this strain can increase the Hb concentration and O_2_ saturation, and significantly decrease the serum alanine aminotransferase concentration in the blood of endurance horses [3].

### 3.5. The Influence of AW_L.pl135_ AW_L.pc244_ and Their Combination on Newborn Calves’ Feces Microbiological Parameters

The microbiological parameters of the feces of calves, fed solely with milk replacer (C_control_) or supplemented with AW_Lpl135_, AW_Lc244_, and AW_L.pl135 × L.pc244_, are presented in Table 6. Results of the ANOVA test indicate that there is an interactive effect (*p* < 0.0001) between the feeding duration and feed supplement used on the microbiological parameters of the calves’ feces. It was found that the C-AW_Lpl135_ and C-AW_Lc244_ groups’ feces had significantly lower *Enterobacteria* counts (lower on average by 45.8 and 4.90%, respectively) on day 14, in comparison with the control group. In addition, the C-AW_Lpl135_, C-AW_Lc244_, and C-AW_L.pl135 × L.pc244_ groups’ feces had significantly higher LAB numbers (higher on average by 5.2, 12.6, and 4.5%, respectively) on day 14, in comparison with C_control_ group feces. After 14 d of supplement feeding, significantly higher YM counts in the C-AW_Lpl135_, and C-AW_Lc244_, groups’ feces samples, by 31.6, and 38.1%, respectively were found. However, in the C-AW_L.pl135 × L.pc244_ group, after 14d of supplement feeding, counts lower by 14.2% YM were found. In the C_control_ group, YM count increased by 17.7% after 14 d. of treatment. These results could be explained, by the fact, that LAB and yeast have a synergistic interaction [110,111]. However, most of the mold’s growth can be suppressed by LAB, and this antifungal activity is highly preferable in order to reduce fungal counts in the digestive tracts of farm animals [111].

It has been suggested that *Lp. plantarum* feed supplement significantly increases the number of LAB in calves’ feces and can improve the balance of enteric microbial flora of these farm animals [112]. According to Fan et al. [113], *Limosilactobacillus reuteri*, *Lactobacillus johnsonii*, *Lactobacillus amylovorus*, and *Ligilactobacillus animalis* inhibit pathogenic strains, including *E. coli* K88 and *S. typhimurium*. These findings indicate the importance of a diverse gut microbiota in respect of newborn calves’ health, due to suppressing pathogen colonization in the gastrointestinal tract to prevent calf diarrhea. Similar tendencies were reported by Vadopolas et al. [1], who suggested that significant accrual of LAB in newborn calves’ feces can be observed after 14 days of feeding with a supplement containing *Liq. uvarum* [1]. Furthermore, calves treated with probiotic LAB strains had lower fecal counts of *Clostridium* spp., higher counts of *enterococci*, and moderate changes in the counts of *Faecalibacterium, Bifidobacterium*, and *Bacillus* spp. [114]. In addition, feed supplements based on *lactobacilli* and administered for 14 days in a row have an affirmative effect on microbial populations in the intestinal tract of calves, leading to higher counts of *Lactobacillus* and *Bifidobacterium* [1,115]. It is known that the administration of lactobacilli and *Bifidobacterium* has a strong and positive impact on animal and human health [116,117]. However, the nutritional manipulations of the rumen microbiome to enhance productivity and health are rather limited due to the resilience of the ecosystem once it has been established in the mature rumen. It has been suggested that the microbial colonization that occurs soon after birth can open the possibility of manipulation with the potential to produce lasting effects into adult life [118,119]. 

In our study, in most cases YM counts increased after 14 days of supplement feeding, and these results could be explained by the fact, that LAB and yeast have a synergistic interaction [110]. However, most of the mold’s growth can be suppressed by LAB, and this antifungal activity is highly preferable in order to reduce fungal counts in the digestive tracts of farm animals [111]. Molds can generate not only allergenic spores, but also secondary metabolites such as mycotoxins, which can cause essential health hazards, not only for livestock, but through production to the final consumer. For this reason, it is very important to avoid their high numbers in farm environments [21,111].

## 4. Conclusions 

The results of this study show that *Lactiplantibacillus plantarum* LUHS135 (Lpl135), *Lacticaseibacillus paracasei* LUHS244 (L.pc244), and their combination possess desirable antimicrobial properties, as well as being non-resistant to all of the tested antibiotics. Additionally, 3% concentrations of L.pl135 and L.pc244 reduce AFB1, OTA, ZEA, T-2, and HT-2 mycotoxin concentrations. Moreover, AW fermented for 48 h with L.pl135 and L.pc244 strains could be recommended as a feed supplement, due to viable LAB cell counts higher than 7.00 log_10_ CFU mL^−1^ and L(+) lactic acid concentrations (>4.5 g L^−1^) with pH values lower than 4.03. AW fermented with L.pl135, L.pc244, and their combination can be used on farms for feeding newborn calves due to their positive effects on certain health parameters, such as reducing the risk of developing acidosis, decreasing the *Enterobacteriaceae* count, and increasing the LAB count in the feces of newborn calves. Finally, valuable feed supplements can be prepared in a sustainable manner by applying the valorization of dairy industry by-products. However, further examination with considerable cohorts of calves is required to confirm these data and also to clarify the mechanism of action of the tested formulations.

## Figures and Tables

**Figure 1 animals-13-03345-f001:**
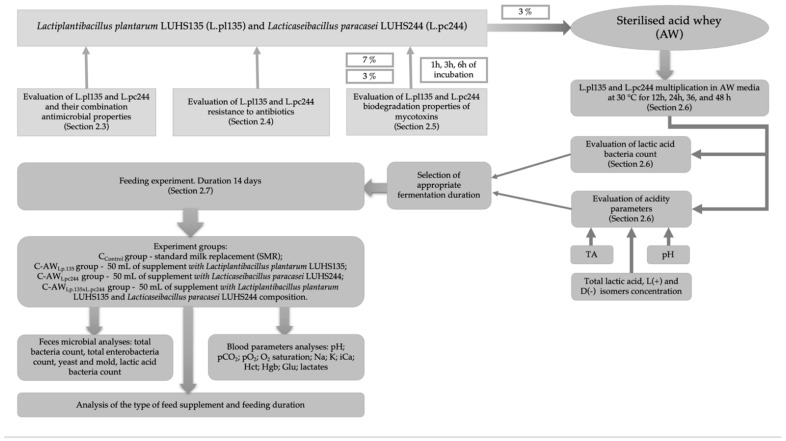
Principal scheme of the experiment. AW—acid whey; L.pl135—*Lactiplantibacillus plantarum* LUHS135; L.pc244—*Lacticaseibacillus paracasei* LUHS244; C—calves; TA—titratable acidity; pH—potential of hydrogen; pCO_2_—partial pressure of carbon dioxide; pO_2_—partial pressure of oxygen; O_2_ saturation—oxygen saturation; Na—sodium; K—potassium; iC—ionized calcium; tCO_2_—total carbon dioxide; Hct—hematocrit; Hgb—hemoglobin content; Glu—glucose; Lac—lactate.

**Figure 2 animals-13-03345-f002:**
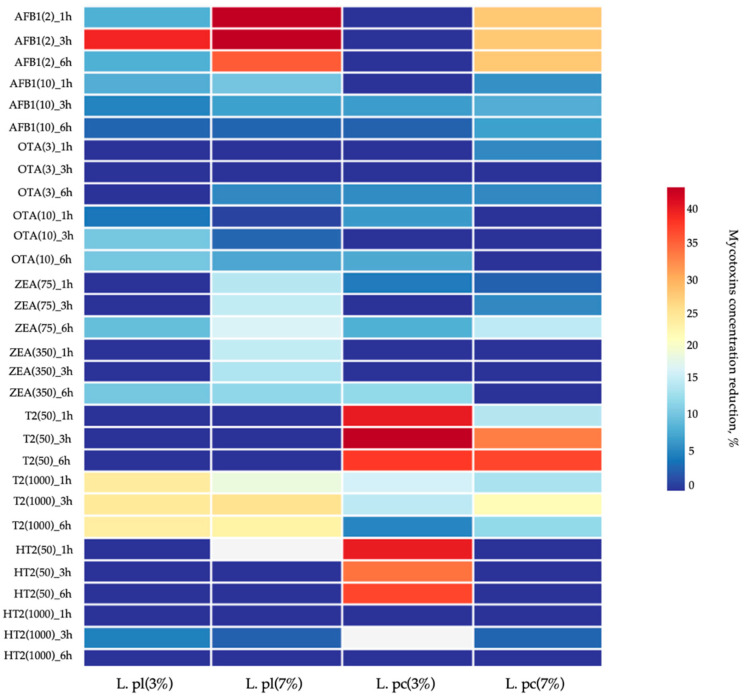
Mycotoxin concentration reduction (%) in mediums after 1 h, 3 h, and 6 h of incubation. AFB1—aflatoxin B1(samples were spiked with 2.00 and 10.0 μg L^−1^ concentrations of AFB1); OTA—ochratoxin A (samples were spiked with 3.00 and 10.0 μg L^−1^ concentrations of OTA); ZEA—zearalenone (samples were spiked with 75.00 and 350.0 μg L^−1^ concentrations of ZEA); T-2—T-2 toxin (samples were spiked with 50.0 and 1000.0 μg L^−1^ concentrations of T-2); HT-2—HT-2 toxin (samples were spiked with 50.0 and 1000.0 μg L^−1^ concentrations of HT-2); L.pl135 (*Lactiplantibacillus plantarum* LUHS135); L.pc244 (*Lactocaseibacillus paracasei* LUHS244).

**Figure 3 animals-13-03345-f003:**
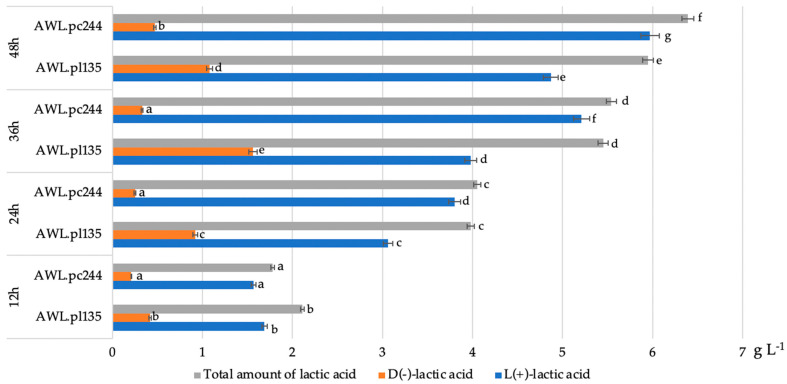
Lactic acid and L(+) and D(−) lactic acid isomer concentrations in acid whey after different fermentation times. ^a–g^ different letters indicate significant time-related differences (within the samples fermented with both LAB strains), *p* ≤ 0.05); data are expressed as the mean values (*n* = 3) ± standard error (SE). AW—acid whey; L.pl135—*Lactiplantibacillus plantarum* LUHS135; L.pc244—*Lactocaseibacillus paracasei* LUHS244.

**Table 1 animals-13-03345-t001:** Experimental diet feeding composition for newborn calves.

Experimental Groups	Experimental Diet Feeding Composition				
	SMR, L Per Calf Per Day	AW, mL	L.pl135, log_10_ CFU mL^−1^	Lpc244, log_10_ CFU mL^−1^	L.pl135 × Lpc244, log_10_ CFU mL^−1^
C_control)_	8–10	50	-	-	-
C-AW_L.pl135_			>7.0	-	-
C-AW_Lpc244_			-	>7.0	-
C-AW_L.pl135×Lpc244_			-	-	>7.0

C—calves; AW—acid whey; L.pl135—*Lactiplantibacillus plantarum* LUHS135; L.pc244—*Lactocaseibacillus paracasei* LUHS244; SMR—standard milk replacer; CFU—colony forming units.

**Table 2 animals-13-03345-t002:** *Lactiplantibacillus plantarum* LUHS135 and *Lactocaseibacillus paracasei* LUHS244 strains’ antimicrobial activity.

Pathogenic and Opportunistic Strain	Diameter of Inhibition Zone, mm		
	L.pl135	L.pc244	L.pl135 × L.pc244
*Pseudomona aeruginosa*	13.5 ± 0.2d	12.0 ± 0.3b	13.1 ± 0.4
*Staphylococcus aureus*	13.0 ± 0.4d	13.5 ± 0.3c	12.4 ± 0.3e
*Escherichia coli*	10.5 ± 0.5a	10.5 ± 0.4a	9.00 ± 0.1a
*Streptococcus mutans*	15.5 ± 0.2e	14.5 ± 0.7c	12.8 ± 0.5e
*Enterococcus faecium*	14.2 ± 0.3d	14.0 ± 0.4c	10.4 ± 0.4c
*Klebsiella pneumoniae*	12.0 ± 0.1c	13.5 ± 0.6c	nd.
*Enterococcus faecalis*	11.5 ± 0.3b	11.0 ± 0.2a	9.4 ± 0.2b
*Bacillus cereus*	17.0 ± 0.6f	14.5 ± 0.5c	15.5 ± 0.7
*Proteus mirabilis*	14.0 ±0.4d	13.5 ±0.4c	12.4 ± 0.4e
*Citrobacter freundii*	13.7 ± 0.5d	16.0 ± 0.5d	17.8 ± 0.6f
*Salmonella enterica*	10.5 ± 0.2a	10.5 ± 0.5a	11.3 ± 0.3d

Data expressed as a mean value (*n* = 3) ± SE; SE—standard error. a–f Mean values with different superscript letters within the same LAB strain are significantly different (*p* ≤ 0.05); L.pl135—*Lactiplantibacillus plantarum* LUHS135; L.pc244—*Lactocaseibacillus paracasei* LUHS244. L.pl135 and L.pc244 strains were considered non-resistant to all EFSA-recommended antibiotics when the MIC (mg mL−1) values obtained were the same or lower than the recommended breakpoint value, defined at the species level by the FEEDAP (Panel on Additives and Products or Substances used in Animal Feed Breakpoint) [43] (Table 3).

**Table 3 animals-13-03345-t003:** *Lactiplantibacillus plantarum* LUHS135 and *Lactocaseibacillus paracasei* LUHS244 strains’ resistance to antibiotics.

Antibiotics	L.pl135	L.pc244	FEEDAP Breakpoint, mg mL ^−1^
	Resistance to Antibiotics, MIC, mg mL^−1^		
GEN	16.0	16.0	16
TET	1.50	1.00	8
ERY	0.250	0.500	1
AML	0.047	0.032	n.r.
TMP	0.25	0.75	n.r.

L.pl135—*Lactiplantibacillus plantarum* LUHS135; L.pc244—*Lactocaseibacillus paracasei* LUHS244; GEN—gentamicin; TET—tetracycline; ERY—erythromycin; AML—amoxicillin; TMP—trimethoprim; MIC—minimum inhibitory concentration; n.r.—not required; nd.—not detected; FEEDAP—Panel on Additives and Products or Substances used in Animal Feed Breakpoint [43].

**Table 4 animals-13-03345-t004:** Lactic acid bacteria (LAB) count and acidity parameters (pH and total titratable acidity) of fermented acid whey.

	Duration of Fermentation			
	12 h	24 h	36 h	48 h
	LAB Count log_10_ CFU mL^−1^			
AW_L.pl135_	6.29 ± 0.09^b;A^	6.89 ± 0.05^b;B^	7.14 ± 0.06^a;C^	7.53 ± 0.12^a;D^
AW_L.pc244_	5.20 ± 0.11^a;A^	6.07 ± 0.12^a;B^	7.00 ± 0.10^a;C^	7.48 ± 0.08^a;D^
	pH			
AW_L.pl135_	4.45 ± 0.02^a;D^	4.05 ± 0.03^a;C^	3.97 ± 0.02^a;B^	3.89 ± 0.02^a;A^
AW_L.pc244_	4.64 ± 0.01^b;C^	4.28 ± 0.02^b;B^	4.07 ± 0.03^b;A^	4.03 ± 0.01^b;A^
	TA, °N			
AW_L.pl135_	6.1 ± 0.1^b;A^	6.4 ± 0.1^a;B^	7.4 ± 0.2^b;C^	7.8 ± 0.1^b;D^
AW_L.pc244_	5.0 ± 0.1^a;A^	6.2 ± 0.1^a;B^	6.5 ± 0.1^a;C^	6.9 ± 0.1^a;D^

^A–D^ different capital letters indicate significant time-related differences (*p* ≤ 0.05);.^a,b^ different letters indicate differences among lactic acid bacterial strains used (*p* ≤ 0.05); data are expressed as the mean values (*n* = 3) ± standard error (SE). AW—acid whey; L.pl135—*Lactiplantibacillus plantarum* LUHS135; L.pc244—*Lactocaseibacillus paracasei* LUHS244; LAB—lactic acid bacteria; TA—titratable acidity.

**Table 5 animals-13-03345-t005:** Blood parameters of calves fed with milk replacer only, or supplemented with AW_Lpl135_, AW_Lc244_, and their combination—AW_L.pl135 × L.pc244_.

Variable	Day	Treatments				*p*-Value
		C_control_	C-AW_L.pl135_	C-AW_L.pl135_	C-AW_L.pl135 × L.pc244_	Day × Treat Int
pH	Baseline	7.32 ± 0.04^A;a^	7.36 ± 0.03^A;a^	7.37 ± 0.01^A;a^	7.31 ± 0.06^A;a^	0.457
	14	7.36 ± 0.02^A;a^	7.38 ± 0.01^A;a^	7.38 ± 0.03^A;a^	7.38 ± 0.04^A;a^	
pCO_2_, mmHg	Baseline	66.73 ± 3.29^B;a^	58.73 ± 2.16^B;a^	61.72 ± 3.51^B;a^	67.03 ± 3.09^A;a^	0.0001
	14	57.35 ± 1.49^A;c^	32.15 ± 1.85^A;a^	37.70 ± 2.11^A;b^	63.85 ± 3.18^A;d^	
pO_2_, mmHg	Baseline	19.20 ± 1.61^A;a^	22.70 ± 1.01^A;a^	21.11 ± 1.65^A;a^	26.46 ± 1.53^A;b^	0.0001
	14	18.90 ± 1.91^A;a^	24.90 ± 1.55^A;b^	37.70 ± 1.11^B;c^	41.40 ±2.10^B;d^	
O_2_ saturation, %	Baseline	7.97 ± 0.44^A;a^	8.07 ± 0.17^B;a^	10.80 ± 0.87^A;b^	19.18 ± 1.02^A;c^	0.079
	14	7.35 ± 0.63^A;b^	7.01 ± 0.19^A;a^	11.10 ± 0.93^A;c^	20.35 ± 1.02^A;d^	
Na, mmol L^−1^	Baseline	137.24 ± 2.62^A;a^	136.36 ± 2.83^A;a^	141.28 ± 2.52^A;a^	138.25 ± 3.77^A;a^	0.319
	14	135.74 ± 1.71^A;a^	136.16 ± 3.12^A;a^	135.27 ± 2.52^A;a^	134.25 ± 3.02^A;a^	
K, mmol L^−1^	Baseline	5.11 ± 0.16^B;a^	5.07 ± 0.27^A;a^	4.97 ± 0.15^A;a^	5.38 ± 0.46^A;b^	0.048
	14	4.45 ± 0.24^A;a^	5.25 ± 0.15^A;b^	4.67 ± 0.25^A;a^	4.70 ± 0.35^A;a^	
iCa, mmol L^−1^	Baseline	1.33 ± 0.03^A;a^	1.25 ± 0.06^A;a^	1.23 ± 0.03^A;a^	1.23 ± 0.03^A;a^	0.001
	14	1.40 ± 0.02^B;b^	1.40 ± 0.02^B;b^	1.20 ± 0.02^A;a^	1.18 ± 0.06^A;a^	
tCO_2_, mmHg	Baseline	36.17 ± 1.98^A;a^	35.23 ± 2.46^A;a^	37.90 ± 2.12^A;a^	42.85 ± 2.73^A;a^	0.458
	14	34.40 ± 1.41^A;a^	33.90 ± 1.19^A;a^	38.10 ± 1.65^A;a^	39.40 ± 1.23^A;a^	
Hct, % fraction	Baseline	20.30 ± 1.81^A;a^	22.00 ± 1.97^A;a^	30.00 ± 1.46^A;b^	29.25 ± 1.30^B;b^	0.454
	14	17.02 ± 1.41^A;a^	25.53 ± 1.51^A;b^	31.30 ± 1.08^A;c^	24.25 ± 1.53^A;b^	
Hgb, g dL^−1^	Baseline	6.92 ± 0.32^A;a^	7.50 ± 0.56^A;b^	10.30 ± 0.21^A;d^	9.95 ± 0.64^A;b^	0.0001
	14	5.71 ± 0.56^A;a^	8.72 ± 0.62^A;b^	10.53 ± 0.34^A;c^	13.68 ± 0.84^B;d^	
Glu, mmol/L	Baseline	5.67 ± 0.13^A;c^	4.80 ± 0.24^A;a^	6.83 ± 0.36^B;d^	4.93 ± 0.27^A;b^	0.517
	14	5.25 ± 0.49^A;b^	5.37 ± 0.35^A;b^	5.37 ± 0.48^A;b^	4.30 ± 0.37^A;a^	
Lactate, mmol/L	Baseline	4.22 ± 0.09^B;d^	2.30 ± 0.09^B;b^	2.10 ± 0.08^B;a^	3.40 ± 0.03^B;c^	0.0001
	14	3.66 ± 0.04^A;d^	1.33 ± 0.03^A;a^	1.44 ± 0.05^A;b^	1.56 ± 0.07^A;c^	

Data are presented as mean ± SE (*n* = 12/group); baseline measurements were performed on day 2, before the start of the feeding experiment; ^A,B^ different capital letters indicate significant time-related differences (*p* ≤ 0.05); ^a–d^ different letters indicate differences among treatments (*p* ≤ 0.05). C—calves; AW—acid whey; L.pl135—*Lactiplantibacillus plantarum* LUHS135; L.pc244—*Lactocaseibacillus paracasei* LUHS244.pCO_2_ —carbon dioxide partial pressure; pO_2_—partial pressure of oxygen; iCa—ionized calcium; Glu—glucose, Hct—hematocrit fraction; tCO_2_—total amount of CO_2_; Hgb—hemoglobin content.

**Table 6 animals-13-03345-t006:** Microbiological parameters of calves’ feces.

Variable	Day	Treatments				*p*-Value
		C_control_	C-AW_L.pl135_	C-AW_L.pc244_	C-AW_L.pl135 × L.pc244_	Day × Treat Int
TCM	Baseline	7.09 ± 0.02^B;a^	7.71 ± 0.05^B;b^	7.11 ± 0.09^B;a^	7.71 ± 0.04^B;b^	0.0001
	14	7.94 ± 0.05^A;c^	6.87 ± 0.07^A;a^	6.83 ± 0.04^A;a^	7.51 ± 0.06^A;b^	
LAB	Baseline	6.16 ± 0.04^A;b^	6.02 ± 0.08^A;b^	6.25 ± 0.04^A;c^	5.37 ± 0.05^A;a^	0.0001
	14	6.19 ± 0.09^A;a^	6.53 ± 0.09^B;b^	7.08 ± 0.06^B;c^	6.45 ± 0.08^B;b^	
TCE	Baseline	7.11 ± 0.07^B;a^	7.57 ± 0.04^B;c^	7.13 ± 0.03^B;a^	7.38 ± 0.08^B;b^	0.0001
	14	6.80 ± 0.09^A;c^	4.77 ± 0.07^A;a^	6.47 ± 0.09^A;b^	7.05 ± 0.06^A;d^	
YM	Baseline	2.03 ± 0.05^A;a^	3.19 ± 0.08^A;c^	2.91 ± 0.02^A;b^	5.58 ± 0.03^B;d^	0.0001
	14	2.39 ± 0.06^B;a^	4.20 ± 0.06^B;c^	4.02 ± 0.03^B;b^	4.79 ± 0.07^A;d^	

Data are presented as mean ± SE (*n* = 12/group). Baseline measurements were performed on day 2, before the start of the feeding experiment. ^A,B^ different capital letters indicate significant time-related differences (*p* ≤ 0.05); ^a–d^ different letters indicate differences among treatments (*p* ≤ 0.05); C—calves; AW—acid whey; L.pl135—*Lactiplantibacillus plantarum* LUHS135; L.pc244—*Lactocaseibacillus paracasei* LUHS244; TCM—total count of aerobic and facultative anaerobic microorganisms; LAB—lactic acid bacteria count; TCE—total count of enterobacteria; YM—yeasts/molds.

## Data Availability

Not applicable.

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
