# Peer review of "Antimicrobial and Mycotoxin Reducing Properties of Lactic Acid Bacteria and Their Influence on Blood and Feces Parameters of Newborn Calves"

_animals, 2023, doi:10.3390/ani13213345_

Round 1
Reviewer 1 Report
Comments and Suggestions for Authors
The present work was well appreciated and appropriate for this journal. However, several blemishes need revision before this work can be considered.
Abbreviations must be defined at first mention and used consistently thereafter in the entire manuscript. Revise the entire manuscript. Ex. Line no. 133; 311-315;
Include one of the tables for the Ingredient composition, of experimental diets.
Why authors not obtain the growth performance results?
Your manuscript would greatly improve from editing by a native English speaker.
In vitro – should be in italics
Microbial strains – must be in italics. Line no.: 137-139
Suggested to avoid multiple citations for single sentence Ex. Line no.: 63, 97... If necessary means explain them individually.
Suggested to improve the quality of graph figures.
Comments on the Quality of English LanguageYour manuscript would greatly improve from proofread editing by a native English speaker.
Author Response
The authors are sincerely grateful for the comments. Changes and answers to comments are given in the manuscript and attached file.

Reviewer 2 Report
Comments and Suggestions for Authors
Dear editor and the authors;
I have reviewed the manuscript titled “Application of Antimicrobial and Mycotoxins Reducing Properties Possessing Lactic Acid Bacteria Biomass for Newborn 3 Calves Feeding”. The study describes the application of direct-fed microbials from two different lactic acid bacteria strains and their combinations in terms of in vitro antimicrobial resistance, inhibition of growth of harmful bacteria and reduction of several mycotoxins. On the other hand, a field study was conducted to evaluate the changes in health parameters of Holstein calves whose diets were supplemented with the above mentioned DFMs.
I believe the provided results are important for animal wellbeing and have a potential to be extended to commercial applications. However, before that, more in vivo tests should be conducted, especially on the effect of the DFM on pathogenic microorganisms in cattle feces and reduction of mycotoxins in milk.
In general, I would like to support the publication of this manuscript, however, some issues should be addressed before acceptance. In general, the discussion should be improved, minor language editing is needed and some statistical analysis should be revisited.
My specific comments are given below:
Abstract: Some abbreviations first appear in the “Simple summary” section, therefore, not spelled out in the abstract. However, most databases (such as PubMed) will only include “abstract”, therefore I think the abbreviations should also be defined in the abstract section. Please consult with the editorial team to learn about an appropriate way of including abbreviations in the abstract.
1. Introduction
Lines 83-84: A major concern about AFB1 contamination in feed is that when dairy cows are fed, AFM1 (a toxic metabolite of AFB1) is excreted through their milk, therefore posing risks for human health as well. Please also include this information. It is particularly important for this study as the target animal group is Holstein.
Line 92: What is the relationship between mycotoxin degradation and proteolytic activity? Most mycotoxins are not peptides and enzymes responsible for mycotoxin degradation are mostly oxidases or peroxidases. Suggested reading: https://doi.org/10.3390%2Ftoxins9040111.
Lines 103-104: Research and commercial applications of DFMs were previously reported extensively, so the application might not be that much limited. Suggested reading: https://doi.org/10.1111/zph.12112 and there are also other primary research and review papers. Bovamine Defend is a well established example of an LAB DFM used for cattle. Probicon L28 is a novel one for cattle and Bioplus 2B is for swine. Although none of these were targeted for mycotoxins, evaluation of effectiveness against foodborne pathogens (especially 0157:H7) is well documented and should be mentioned in this manuscript.
I think it would be useful to talk about the commercialization opportunities for this novel DFM briefly.
2. Materials and methods
In general, number of samples is missing in the M&M section.
Lines 133-136: Please indicate if the AW resulted from food processing (yogurt or cheese?) or it was acidified for any other reason.
Lines 137-138: Organism names should be typed in italic. Also the authors need to spell out the genus names, as most of them did not appear in the text previously. E.g. S. aureus -> Staphylococcus aureus. Please also include more detail about the stains used, especially if they have common identifying numbers (such as ATCC code).e.g. E. coli is very broad, was it generic or pathogenic? Were these strains previously related to any outbreaks or isolated from farm animals?
Line 142: “TM” may also refer to Tobramycin, please use a common abbreviation convention for antibiotics. Please also check others. (suggested use: https://journals.asm.org/abbreviations-conventions or if this journal has any requirements)
Lines 152-154: Please indicate these tests are in vitro
Lines 156-159: I wonder why such low concentrations were used. For example, aflatoxin limits are usually around 20 ppb for animal feed, therefore it would be more useful to test concentrations at or above these limits. Please justify this selection.
Figure 1: The figure is nice, however quite hard to follow. I suggest referring to relevant sections in M&M. E.g., please indicate “Evaluation of resistance to antibiotics” refers to Section 2.3 and so on.
I don’t think 3 v/v and 7 v/v are correct, they should be percentages.
Some of the arrows seem weird. Please use a better chart drawing tool (Microsoft Visio, draw.io, etc.)
Line 183: It is not clear what this “substance” is. Was a culture of the target LAB directly placed into the well or were there any downstream processes?
Line 196: “Contamination” generally refers to unintentional instances. I think better wording would be “challenged” or “spiked”.
Line 204: Mycotoxin extraction methods should also be mentioned in this section. There are different complex methods for extraction that can affect the subsequent analyses, therefore affect the replicability of the methods given in the manuscript.
Line 211: How was AW sterilized?
Lines 212-214: How was the fermentation done? If in flask, what was the ratio of the container to the liquid, any agitation?
Lines 214-215: How were these samples taken without breaking the anaerobic seal? Were these independent samples or was there a mechanism to take samples from fermentation containers without breaking the seal?
Lines 223-224: Please cite those previously described procedures.
Section 2.7: As much information available as possible should be given about the animals. Age, body weight for each group, previous diet, housing conditions and any other variables.
3. Results and Discussion
Line 263: Please correct “photogenic”.
Table 1: I think this table needs to be split as it reports information from two different experiments and presenting it like this will lead to misunderstanding. One might ask what the relationship between GEN is and Pseuromonas, E. coli & Streprococcus. I guess they are not related.
Footnotes for this table are given as a separate row, which I think should be given outside of the table.
Lines 305-307: Please also note that you don’t provide any results about the presence of resistance genes, which is outside the scope of this study.
Lines 313-314: Abbreviations for mycotoxins [aflatoxin B1(AFB1), ochratoxin A 313 (OTA), zearalenone (ZEA), T-2 toxin (T2), and HT-2 toxin (HT2)] were defined previously, no need to give them in parentheses again, you can just provide the abbreviations.
Lines 337-335: When giving reductions, please also provide corresponding statistical measures (p-values). (Same applies to other sections, i.e. OTA, ZEA and T-2)
Figure 2 (and all other plots): These plots are not publication quality. Please revise them, preferably use a software tool designed for scientific publications (e.g. SigmaPlot). Decimals are given by commas, need to be consistent with the text.
Line 339: I don’t think “treated with” is the correct term here. These samples were challenged, not treated. The treatment would be the mechanism of action that the LAB strains provide. In the text, you mention that the samples were treated with the strains, not the mycotoxins (Please see Line 378 for example)
In this plot, we also see some increases in AFB 1 concentrations, that are outside the statistical confidence limit, however, discussion of these observations is missing. (Also for OTA, ZEA and T-2)
Lines 343-344: I think this information is incomplete. There can be other mechanisms of AFB1 reduction, such as enzymatic degradation. (Please use search term “aflatoxin oxidase”).
Line 349: Delete “In comparison”. In comparison with what?
Lines 409-410: How are your findings in agreement with the results from these studies? As far as the methodology section is concerned, I don’t see that the authors tested if the ZEA reduction is the result of an adsorption process.
Line 411: In comparison with what?
Line 417: “Opposite tendencies”, opposite of what?
Lines 476-477: It is not appropriate to report increase in cell numbers as percentages. Use logarithmic increase instead.
Lines 480-481: It is not appropriate to express the effect of pH as a correlation. If the authors want to study this, they need to calculate the growth rates at different pH values and use a secondary model to discuss the effect of pH on growth rates.
Lines 483-485: In general, pH, water activity and temperature are factors that affect growth. In addition, inhibitory compounds and substrate limitations can be considered from a predictive microbiology point of view.
Lines 492-493: The information about correlation is redundant. As the pH decreases, acidity increases.
Lines 536-539: Not sure if an ordinary ANOVA is enough to correctly analyze the effect of time. When consecutive samples are taken, the previous sampling result is expected to be correlated with the previous sampling results. Therefore a “repeated measures” ANOVA should have been done.
Table 4: Reduction in Enterobacteriaceae for L.pl135 is quite impressive, however, it seems like the combination treatment is a lot less effective in reducing EB than both treatments individually. What might be a reason for this? With that being said, I think more discussion should be given about the counts of TCM, TCE and YM. Given as “yeast/fungi”, however yeast is already fungi. A better naming would be YM:yeasts/molds.
Comments on the Quality of English Language
Overall, the quality of language is fine, however, as the authors are not native speakers, there are some instances where redundant wording could be used. Therefore I suggest a thorough proofreading of the manuscript.
Author Response

(The authors gave the same response as above.)

Reviewer 3 Report
Comments and Suggestions for Authors
Line 235-236 conflicts with Line 247. In Line 235-236, it is stated that the calves were housed in an outdoor box, but Line 247 states that the calves were housed indoors.
Results and Discussion. In each section, clearly present the results before providing any discussion. It is confusing if these are mixed.
Table 1 would be easier to understand if it was divided into two tables, one with the diameters and one with the resistance. Also, this reviewer is skeptical of the reported significant differences shown in Table 1 for Diameter of Inhibition Zones. In several cases, the reported differences among means do not seem to be consistent with the means and deviations. For example, the differences for Streptococcus mutans does not seem to be consistent with expected statistical differences based on the values shown. The authors should verify differences to ensure that the reported differences are accurate.
Figures. Several of the figures are too small to read clearly. Include only one graph or image in each figure and ensure that it is clearly legible. Make sure the base (lowest number) and peak (greatest number) are the same for each graphic in each set of similar graphs. Do not repeat numerical numbers in sentences in the Results and Discussion section if these numbers are clearly shown on a chart or in a table.
Figure 2A and 2B. It is not clear why there was significant variation in among treatments in terms of impact of duration of treatment. Furthermore, most of the reductions were modest. Even though some comparisons were statistically different, was there any true biological impact? At what level would a reduction be significant biologically?
L345 It is not clear what this statement means. Please clarify.
L540 Finally! This last section is the most important part of the entire manuscript, because is reveals the actual beneficial biological effects of specific lactic acid bacteria in young calves. It is this section of the manuscript that makes it worthy of publication. Nevertheless, the manuscript needs to be reduced in scope to focus on the most important findings. Too much of the manuscript is devoted to in vitro studies that cannot be interpreted with any biological basis. Too many of the graphs of in vitro studies show no effects or modest effects, and it is virtually impossible to interpret these data because there is no clear end point. The studies with calves show clear effects.
My overall recommendation would be to reduce the manuscript to the studies with calves, because the in vitro studies are essentially meaningless. The authors did a lot of laboratory work, but those that will read this manuscript will quit reading before they reach the most important findings, which are the studies with calves. My impression is that if the manuscript retains the in vitro studies, it will rarely be read or cited, but if the calf studies alone are published, it will be a highly cited paper.
Author Response

(The authors gave the same response as above.)
